# OpenReview forum: "Scaling Sparse Feature Circuits For Studying In-Context Learning"
_ICLR.cc/2025/Conference — Submitted to ICLR 2025_

### Official Review · Reviewer_JwLJ · 2024-11-03

**Soundness:** 3
**Presentation:** 2
**Contribution:** 3
**Rating:** 6
**Confidence:** 4

**Summary:**

This paper presents novel techniques for analyzing in-context learning (ICL) mechanisms in large language models using sparse autoencoders. The authors develop a task vector cleaning algorithm and modify sparse feature circuits methodology to work with the Gemma-1 2B model. Their key findings include identifying and validating two critical components of ICL circuits: task-detection and task-execution features. The work demonstrates how these components causally interact to enable ICL capabilities, providing new insights into how language models perform this important function.

**Strengths:**

1. The paper makes significant technical contributions by developing novel methods (task vector cleaning algorithm and modifications to SFC) that enable analyzing in-context learning mechanisms in larger language models like Gemma-1 2B, which is substantially larger than models typically studied at this depth.

2. The work provides meaningful insights into how ICL works by identifying and validating two key circuit components (task-detection and task-execution features) and demonstrating their causal relationships, advancing our understanding of the mechanisms behind this important capability.

3. The methodology is rigorous, with extensive ablation studies, careful experimental design, and thorough validation of findings through steering experiments and causal analysis.

**Weaknesses:**

1. The paper's scope is limited to simple task vector settings and may not capture the full complexity of ICL as used in practice with longer sequences and more open-ended tasks.

2. While the authors acknowledge approximation errors in their interpretations (particularly regarding the role of attention heads vs MLPs), more discussion of the potential implications of these approximations on their conclusions would strengthen the paper.

3. The reproduction of results may be challenging since some key implementation details and hyperparameters are not fully specified, particularly regarding the task vector cleaning algorithm.

4. The paper would benefit from more direct comparisons to existing methods for analyzing ICL mechanisms to better contextualize the advantages of their approach.

**Questions:**

1. How might your findings generalize to more complex ICL scenarios involving longer sequences or open-ended tasks? Have you done any preliminary investigations in this direction?

2. Could you provide more details about the hyperparameter selection process for the task vector cleaning algorithm and its sensitivity to different parameter choices?

3. How do the approximation errors in interpreting the roles of attention heads versus MLPs affect the reliability of your conclusions about the causal relationships between task-detection and task-execution features?

4. Could you elaborate on how your approach compares to other methods for analyzing ICL mechanisms in terms of scalability, interpretability, and accuracy?

---

> ### Author Response · Authors · 2024-11-21
> **Rebuttal by Authors**
>
> Thank you for your thoughtful review and detailed feedback on our paper. Below, we address your points and clarify the changes and additions we have made in response to your comments:
>
> # Comparison with existing methods
> In addition to the paragraph in Related Work summarizing past work on interpreting ICL, we have added a more extensive review in the revised paper as Appendix H. This section compares our method to various existing approaches in cases where comparison is possible and draws some connections to past work. In particular, we find similarities with work on anchor tokens from [Wang et al. 2023](https://aclanthology.org/2023.emnlp-main.609/), find similar findings regarding approximation errors in [Bansal et al. 2022](http://arxiv.org/abs/2212.09095), and highlight our focus on mechanistic interpretability for larger models on real-world discrete tasks.
>
> # Generalization to more complex ICL scenarios
>
> From our experiments with algorithmic tasks, we hypothesize that more complex tasks may involve less interpretable features. For example, features in the *algo_last* task were still somewhat interpretable, as they activated on word repetitions. However, features for tasks like *algo_first* and *algo_second* were less clean and harder to interpret from manual inspection of features of Gemma Scopes – they did not seem to activate on examples where a second instance of an object was selected. Despite this, these features still exhibited noticeable and selective effects as shown by steering experiments, indicating that they play distinct and meaningful roles in the ICL process. Since our findings and investigation methods are considerably more in-depth than prior work on task vectors, we are unsurprised that there are some difficulties further scaling our method. However, we think there are no fundamental limitations with SFC and task features and it is a promising area for future research to address.
>
> # Details on hyperparameter selection and sensitivity
>
> We have conducted $L_1$ coefficient sweeps for multiple models and SAEs, including Gemma-1, Gemma-2 2B (16k and 65k sparse autoencoder widths), and our own Phi-3 sparse autoencoders. These results have been added to the Appendix D.1. They show that our method performs generally well across those models and SAEs, being able to reduce active SAE features by 50-70%, while preserving task vector performance.
>
> # Approximation Errors
>
> We have clarified the limitations section (the paragraph in Section 6: conclusion) to better explain the implications of the approximation errors in our analysis of attention heads versus MLPs. When formulating the experiment, we assumed that attention alone was sufficient to describe the causal link between task-detection and task-execution features. However, our experiments revealed that this link required running the MLP on top of direct attention outputs. Moreover, Figure 6 uses frozen attention patterns with ablations on pre-OV residuals instead of SFC nodes, moving the causal link analysis even further from the SFC subcircuit. Since SFC uses transcoders instead of MLPs, this may also introduce additional approximation errors, as this experiment does not account for transcoder effects and relies solely on MLP outputs.

---

> ### Author Response · Authors · 2024-12-02
> **Gentle reminder that we think we've addressed the concerns**
>
> Hello reviewer JwLJ, **we only have 23 hours left to be able to respond to your questions** and concerns. We think that we've addressed them as detailed in our rebuttal. Let us know if you have any further questions

---

### Official Review · Reviewer_BufG · 2024-11-04

**Soundness:** 3
**Presentation:** 3
**Contribution:** 3
**Rating:** 5
**Confidence:** 4

**Summary:**

This paper studies task features for in-context learning (ICL). By applying SAEs, the authors identify sparse features within the task vector that represent task-specific knowledge and can activate tasks in a zero-shot manner, consistent with prior work on task vectors. To further explore this, the study extends the sparse feature circuits methodology to the larger Gemma-1 2B model for ICL tasks. The results reveal task-detecting features that activate early in prompts, signaling task performance and interacting causally with task-execution features through attention and MLP layers, providing new insights into ICL mechanisms in large models.

**Strengths:**

1. understanding how ICL works is an important problem
2. the paper is structured in a good way
3. the presentation using heatmap is intuitive and shows the effectiveness and exclusiveness of these task features

**Weaknesses:**

1. as the author has mentioned in the limitation section, this study only involves Gemma2-2b model. However, to ensure that the method is general, more model should be considered, especially since Gemma Scope also provides trained SAEs for Gemma2-9B models.
2. Fine-tune cost should be included in the paper to show whether the training requires a lot of resource.
3. Ablation study on L1 regularization during fine-tuning should be included to show what is the optimal L1 value for obtaining meaningful sparse features.

**Questions:**

1. the separation of task detection and task execution features is not clear to me. My understanding is that they are just features in earlier and later layers that both represents the task. Is there any more specific characteristics that defines these two kinds of features?
2. the author mentions that ablating a few hundred nodes can cause significant performance drop. Is this number small? What if only the task detection features are ablated? Removing these nodes will shut down the activation of task execution features?
3. since there are a few activated sparse features for the task vector, what is the functionality of other sparse features that are not task features? If all of them can boost the task, then the contribution of this work will be limited.

---

> ### Author Response · Authors · 2024-11-21
>
> We thank the reviewer for their thoughtful feedback and valuable suggestions. Below, we address the points raised:
>
> # Limited evaluation on additional models
>
> We would like to clarify a potential misunderstanding regarding our experimental setup. This study primarily uses the **Gemma-1 2B** model, and we trained our **own SAEs** specifically for this research. Our research started before the release of SAEs for Gemma-2, which is why our main results use our Gemma-1 SAEs.
> However, we have since expanded our analysis to include $L_1$-regularization coefficient sweeps for SAEs from multiple configurations: our Gemma-1 2B (32k), our Phi-3 (50k), and Gemma-2 2B (16k and 65k), as can be seen in Figures 9-13. Our results show that we can reduce the number of active features in task vectors to a fraction of the original, while maintaining performance across setups. We also provide a positive steering chart for cleaned Gemma 2 2B 16k and 65k SAE features (Appendix F.2) together with some of their max activating examples (Appendix I).
> While we acknowledge the importance of applying our method to larger models such as Gemma-2 9B, these models require more complex infrastructure setups. As a result, they are the target of our future research, while this paper focuses on smaller models.
>
> # Training costs
>
> We calculated the costs for training SAEs on TPUs in Appendix B, coming to a figure of 1 week of TPU v4-8 usage. It is likely training costs would lessen significantly on the new generation of TPUs and specifically the v5-e, which would enable us to save device memory and time spent on inference by performing matrix multiplication with quantized weights.
>
> Task vector cleanup is a comparatively cheap algorithm, completing cleanup for a single task and value of $L_1$ coefficient on one TPU chip in about a minute for models ranging from Gemma 1 2B to Phi 3 Mini.
>
> # $L_1$ regularization sweeps
>
> We have included $L_1$ regularization sweeps across the above setups in the Appendix D.1. These results highlight the optimal $L_1$ coefficient​ values for obtaining meaningful sparse features and show that TVC is able to consistently reduce the number of active task features by at least 50% (and up to 70%), while maintaining task vector performance.
>
> # Separation of task-detection and task-execution features
>
> The distinction between task-detection and task-execution features lies both in their activation timing and pattern, and their causal roles within ICL:
> * **Task-detection** features mainly activate on the **output** tokens in the few-shot example pairs on earlier layers, thus detecting the task.
> * **Task-execution** features activate later in the models, on **arrow** tokens specifically.
>
> These patterns also persist in their max activating examples from the training data. *Executors* activate on the tokens just before the something similar to their task is completed, while *detectors* activate on the tokens that complete this task. That is tokens *and* and *cold* in the *hot and cold* example. We have added max activating examples from the SAE training distribution of those two types of features to make the distinction more clear in Appendix I.
>
> Their relationship is supported by experiments such as those in Figure 6, which show that negative steering with task-detection directions on output tokens disables related task-execution features.
>
> # Ablating sparse features
>
> While ablating a few hundred nodes can significantly impair performance, each individual node typically has a low IE, meaning its removal does not entirely disable functionality. The SFC paper was able to disable mechanisms by ablating less than 50 nodes , which is significantly lower than our numbers. This is likely due to the difference in the size of the model (Gemma 1 2B is 30x larger than Pythia 70M) and restoration effects within the model (Explorations of Self-Repair in Language Models by Rushing and Nanda, 2024).
> The steering experiment in Figure 6 demonstrates that ablating task-detection directions disables related task-execution features, although this is not a node-based analysis.
>
> # Functionality of non-task-specific features in task vectors
>
> Many features in task vectors activate on generic patterns commonly associated with ICL, such as arrow tokens or repeated commas. These generic features are less specialized and do not encode the task itself, while still boosting the task vector performance. To illustrate this, we have added full positive steering heatmaps to the appendix (Appendix F), showing that extra features beyond task executors often have broad, unspecific effects.

---

> ### Comment · Reviewer_BufG · 2024-11-25
>
> Thank you for the response and the efforts made in revising the manuscript. Below, I provide my updated comments on the work.
>
> **Evaluation on additional models**:
> I appreciate the inclusion of results for Gemma2 and the exploration of SAE width variations, as suggested by reviewer VLpn. However, I disagree with the assertion that evaluating Gemma2-9B requires significant infrastructure setup. Given that v4 TPUs were utilized in this work, the computational resources should be sufficient for running experiments on Gemma2-9B. Since SEA training is not required and the vector clean-up algorithm is computationally inexpensive (as mentioned in the rebuttal), this evaluation should be feasible.
>
> **$L_1$ regularization sweep**:
> It would be valuable to assess how performance varies when the SAE is trained under different sparsity constraints. As Gemma Scope provides SAEs trained with varying $L_1$ regularization strengths, this experiment would not require training SAEs from scratch, making it relatively straightforward to conduct.
>
> **Separation of task-detection and task-execution features**:
> Thank you for addressing this point and revising Line 407. This clarification highlights an essential distinction between task-detection and task-execution features. But I do think this additional statement is important to distinguish these two features. To further strengthen the argument, empirical evidence should demonstrate that these two feature types **exclusively** activate for their respective input types.
>
> **Ablating sparse features** and **Functionality of non-task-specific features in task vectors**:
> Thank you for addressing my earlier concerns in the revision. The response provided has adequately resolved my questions on these topics.
>
> **Additional consideration**:
> After reviewing the revised manuscript and considering feedback from other reviewers, I believe the paper requires significant restructuring. Several critical details currently relegated to the appendix should be included in the main text to enhance the paper's coherence and readability.
>
> Reviewer JwLJ also raised the important point regarding the roles of MLPs and attention heads. While this has been acknowledged as a limitation in the revised manuscript, I believe it is crucial to address this issue directly in the experiments. Open SAEs like Gemma Scope already include SAEs for both MLPs and attention modules, so conducting such experiments should not be resource-intensive.
>
> Based on the response, the revisions, and feedback from other reviewers, I would like to adjust my rating to 5 and suggest for a complete revision of the work to address these points and improve its structure and presentation

---

> > ### Author Response · Authors · 2024-11-25
> >
> > Dear Reviewer BufG,
> >
> > Thank you for your thoughtful feedback on our revision and the detailed suggestions for improving the manuscript. We appreciate your careful consideration of our work and would like to address your points:
> >
> > **Regarding Gemma2-9B Infrastructure:** While we have access to v4 TPUs, running Gemma2-9B experiments presents engineering challenges beyond raw computational resources. The model needs sharded computation across multiple TPU cores, requiring refactoring of our current JAX codebase to support distributed computing. We think that this engineering effort may extend beyond the rebuttal period.
> >
> > **Regarding Task Detection/Execution Feature Evidence:** We appreciate your suggestion about strengthening the empirical evidence for the distinction between task-detection and task-execution features. We would be grateful if you could clarify what specific types of empirical evidence you would find most convincing. Would including the max activating examples from the training distribution in the main paper (currently in Appendix I) help demonstrate this distinction more clearly?
> >
> > **Regarding Approximation Errors (Response to the Discussion with Reviewer JwLJ):** We want to clarify that the approximation errors discussed with Reviewer JwLJ do not affect the evidence for the detector-executor link. We mentioned these limitations only because we study the link at the model component level (attention heads and MLPs) rather than intermediate SFC transcoder nodes and attention outputs, although we initially discovered it through SFC subcircuit analysis. Given that this limitations section appears to be causing confusion, we are considering removing it in the revised manuscript to improve clarity. We currently don’t see which MLP and attention output SAE experiments may strengthen this part of the paper.
> >
> > **Regarding Paper Restructuring:** We appreciate your suggestion about restructuring the paper and moving critical details from the appendix to the main text. Could you please specify which parts beyond the max activating examples you believe should be moved to the main body? This would help us better understand your vision for improving the paper's structure and presentation.
> >
> > We are committed to improving the manuscript and would greatly appreciate your specific guidance on these points.

---

> ### Author Response · Authors · 2024-11-28
> **Official Comment by Authors**
>
> Thank you again for your feedback. We ran additional experiments and updated the main body of the paper for clarity:
>
> **Gemma 2 9B:** We completed TVC L1 coefficient sweeps with the bigger model by running it with Tensor Parallelism. The canonical Gemma Scope SAEs produced results better than ITO and SAE encoding without major tuning, bu we did need to change the *early stopping steps* parameter from 50 to 200 to achieve higher sparsity by letting the TVC algorithm run for longer. We have considered SAEs of width 16k and 65k, as for our Gemma 2 2B experiments. The new results are in Figure 15.
>
> **Task Detector and Task Executor Feature evidence:** To test the extent to which task executor and task detector features activate on their hypothesized token types in distribution, we computed the sum of the activation values of those features on each of the token types and found the percentage of this *activation mass* corresponding to each of the token types. The results are presented in Tables 1 and 2. They show that over **96%** of the activation mass of task **detector** features corresponds to output tokens, and almost **90%** of the activation mass of task **executor** features is in arrow tokens, which is entirely in line with our expectations. From our results, the evidence is clearer for task detectors being distinct than for task executors, but it is still strong for both types of features.
>
> Less than 1% of the activation mass of task detector features is on arrow tokens. However, around 5% of the mass of task executor features is on output tokens, primarily on translation tasks. This may be due to features that activate weakly on many tokens, even though they have the causal effects of task executor features and activate before task execution, like those seen in Appendix I. Their presence does not detract from the significance of our finding because an overwhelming majority of features follow the expected behavior.
>
> **Restructuring the paper:** We deeply appreciate the reviewer’s valuable feedback regarding the placement of details within the manuscript. To better align with their suggestions, we have revised the paper by moving certain elements from the appendix to the main body. Specifically, we have incorporated the flowchart illustrating the Task Vector Cleanup algorithm as the new Figure 2 and provided additional explanations within the main text. Furthermore, we have included several maximum activating examples from Appendix I in Figure 4 with activation mass distribution tables for detectors and executors (Tables 1 and 2). These additions clearly demonstrate the distinct activation patterns and timing of these feature types. We sincerely hope these adjustments contribute to a more coherent and readable presentation for the esteemed reviewer.
>
> **Regarding SAE training $L_1$ regularization sweeps:** To reiterate, we include experiments sweeping across the *$L_1$ coefficient used in the TVC algorithm* in the Appendix. In the above comment, the reviewer requested sweeps across the *$L_1$ coefficient used for training SAEs*, which is not a parameter of our algorithm and is a detail of the SAE used which may affect quality or sweep performance.
>
> While we acknowledge the importance of the TVC $L_1$ coefficient sweeps, we maintain that they are supplementary to our paper's primary contribution. As noted by Reviewer VLpn, one of our paper's key strengths lies in "the successful implementation of a relatively complex combination of methods" that demonstrates the potential of SAEs. Therefore, while we have included comprehensive L1 sweep results in the appendix, we believe this supporting material is appropriately placed.
>
> Following the feedback, we re-ran our experiments on our experiments on SAEs with varying L1 coefficients (and, as a result, varying average L0 norm) for Gemma 2 2B, recording our results in Figures 13-15. Our method performs well across L0s ranging from 20 to 300, with the only notable takeaway being that very high L0s seem to produce lower L0 task vectors with smaller TVC L1 coefficients.
>
> We would also like to clarify our previous discussion about approximation errors: these do not affect the validity of our detector-executor relationship findings. Our results demonstrate robust causal relationships between these feature types, supported by our expanded empirical evidence.
>
> These revisions maintain the paper's focus on its core contributions while addressing the broader scope of model evaluation and empirical validation requested by the reviewers. We believe these changes have significantly strengthened the manuscript while preserving its essential contributions to understanding ICL mechanisms in large language models. Given the substantial effort to incorporate the reviewers' feedback and enhance the paper's clarity and rigor, we hope these improvements merit a reconsideration of the evaluation score.

---

> ### Author Response · Authors · 2024-12-02
> **Have we addressed your concerns?**
>
> Hello reviewer BufG, **we only have 23 hours left to be able to respond to your questions** and concerns. We think that we've addressed them as detailed in our rebuttal, so could you let us know ASAP about further worries, or otherwise would you be open to revising your amended judgement?

---

> > ### Comment · Reviewer_BufG · 2024-12-03
> >
> > Dear authors,
> >
> > I acknowledge that I have read through your comments and your revision. Based on the discussion, I will keep my current score and suggest for a careful revision that incorporate all feedbacks during the discussion for future submission. Thanks for your effort in the rebuttal!

---

### Official Review · Reviewer_VLpn · 2024-11-04

**Soundness:** 4
**Presentation:** 3
**Contribution:** 3
**Rating:** 6
**Confidence:** 3

**Summary:**

This paper focuses on enhancing the interpretability of large language models (LLMs) through Sparse Autoencoders (SAEs), particularly in understanding the mechanism behind in-context learning (ICL). The authors propose a novel approach that uses SAEs to decompose complex ICL processes into task-specific features, which are responsible for detecting and executing tasks within the model. They identify two key types of features: task-detection features, which recognize tasks early in the prompt, and task-execution features, which are activated to complete the task. By adapting the Sparse Feature Circuits (SFC) methodology to a larger model (Gemma-1 2B), the authors provide a more detailed view of how ICL tasks are processed and executed by language models.

The paper introduces a task vector cleaning (TVC) algorithm that effectively decomposes task vectors into interpretable components using SAEs, leading to improved precision in ICL analysis. By analyzing these sparse task vectors, the authors uncover the causal connections between task-detection and task-execution features, which are mediated through attention layers and multilayer perceptrons (MLPs). This work not only advances the understanding of ICL but also showcases the potential of SAEs as powerful tools for interpretability research in LLMs, paving the way for future studies to refine and control model behavior.

**Strengths:**

One of the strengths of this paper is the successful implementation of a relatively complex combination of methods. From my perspective, the paper effectively combines pretrained SAEs, Inference-Time Optimization (ITO), and end-to-end loss into a cohesive approach. The fact that the authors managed to integrate these techniques and make them work together is a significant contribution, showcasing the potential of this hybrid methodology in enhancing model interpretability.

Another key advantage is that this approach eliminates the need to retrain an SAE. Given that training SAEs is a resource-intensive and time-consuming task, avoiding this step greatly reduces the computational burden. This not only makes the proposed method more practical for widespread use, but also encourages further research and experimentation within the community by lowering the entry barrier for other researchers.

**Weaknesses:**

One of the key weaknesses of this paper is the limited selection of models for experimentation. Currently, several models, including the Gemma-2 series (2B and 9B) and some from Qwen, have released portions of their SAE weights. I believe the authors should explore their method’s performance across a broader range of model architectures and parameter scales. Given the complexity of the proposed approach, there is some doubt as to whether the model would maintain the same level of effectiveness on more intricate SAE architectures. Testing the method on different models would help clarify its robustness and generalizability.

Additionally, the paper does not present a comparative analysis of the proposed method under varying SAE widths. For example, there is no exploration of how the method’s performance, in comparison to SAE reconstruction, shifts with different SAE widths. I believe the authors should provide multiple charts that, similar to Figure 2, compare their method’s stability across different SAE widths. There is a possibility that when the SAE width changes, the advantages of this method might diminish, particularly if the gap in sparsity between the SAE and the proposed method narrows.

Another potential weakness is the lack of detailed case studies. The paper would benefit from more specific examples that demonstrate how intervening on particular features affects task vectors in ICL. For instance, showing how intervention on specific task features leads to changes in ICL outcomes, or how the model loses its ICL capability, would significantly strengthen the paper’s claims and provide deeper insights into the causal mechanisms behind the proposed approach.

**Questions:**

- How well your cleaning method works in different model architecture e.g. Qwen 0.5B?
- How about different SAE widths? i.e. will a wider SAE further benefit your method or in contrast, hurt the performance?
- Is there any explanation to feature 1878 in Figure 3 that has effect to both it_en and algo_second, also feature 11173 ?

---

> ### Author Response · Authors · 2024-11-21
> **Rebuttal by Authors**
>
> We thank the reviewer for their thoughtful and constructive feedback. Below, we address the concerns and questions raised:
>
> # Limited selection of models for experimentation
>
> While a broad comparison of different SAE architectures and models is beyond the scope of this paper, we appreciate the importance of evaluating robustness across diverse settings. In response, we have added an analysis of the $L_1$ coefficient sweeps for optimal task vectors across three models (Phi-3, Gemma-1 2B, and Gemma-2 2B) and two SAE widths (16k and 65k) for Gemma-2 2B (Appendix D.1, Figures 9-12). These results demonstrate that our method consistently reduces the number of active task features by at least 50% (and up to 70%), while maintaining task vector performance. Additionally, our method almost always beats ITO (a fact that is somewhat obscured by all tasks being plotted on the same charts). This suggests that our method generalizes well across different configurations, though we acknowledge that more comprehensive studies could further substantiate this claim.
>
> # Exploration of varying SAE widths
>
> We think that our findings are not greatly impacted by scoping to one SAE width only.
>
> 1. Our central findings are related to the interpretation of ICL in terms of SAE features. Using our 32k width SAEs on Gemma 1 2B, we were able to find interpretable decompositions of comparable quality to the task vectors (Figure 2) and find individual features impacting specific tasks (Figure 3). Therefore our work shows a strong lower bound for how useful SAEs are for explaining ICL, and is successful as a proof-of-concept for interpreting ICL with SAEs, which stands without needing to sweep widths.
>
> 2. We added positive steering experiments for Gemma-2 2B SAEs to Figure 19 in Appendix F.2. These experiments required a more complex setup compared to Gemma-1, as we could not identify an optimal steering scale and had to optimize it at the feature level. Despite this, we found that the task-specificity level of Gemma-2 SAEs was similar to that of Gemma-1 SAEs. One notable difference is that the wider 65k SAEs appear to have a larger amount of effective executors in the cleaned task vectors. However, this may be attributed to the low $L_1$ regularization coefficient.
>
> # Lack of detailed case studies
>
> We agree that detailed case studies are important for understanding the behavior of task features. While an extensive exploration of case studies is beyond the primary scope of this paper, we have included some additional analyses to address this point. Specifically, we provide full unfiltered positive steering plots for Gemma-1 (Appendix F) and Gemma-2 2B SAEs (Appendix F.2), along with max-activating examples of prominent features (Appendix I). Additionally, we include a negative steering chart (Appendix F.1) to demonstrate the limitations of steering-based analysis and emphasize the need for advanced methods like Sparse Feature Circuits (SFC) for a deeper investigation of in-context learning. We hope these additions help to partially address the reviewer’s concerns.
>
> # Questions about specific features
>
> 1. **Feature 11173:** From its max-activating examples, we hypothesize that it represents a "provide information about location" feature, as it activates on phrases like *<some name>, <city of residence>*. Its effect, although very weak, on translation tasks might stem from the fact that geographical names often have non-English origins, and translation task features can be explained as the "meaning of this in another language" features. These connections remain speculative but highlight intriguing areas for further study.
>
> 2. **Feature 1878:** Algorithmic features, like this one, tend to encode less interpretable relationships between objects. This could explain their effects, although weak again, on other tasks, boosting logits of tokens related with the input one.
>
> # Future directions
>
> One of our future research directions is studying task feature arithmetic. For example, we aim to investigate how translation task vectors encode target languages by leveraging features from different foreign languages. We also think it is an exciting future direction to conduct more experiments with algorithmic features, such as exploring function vector arithmetics as introduced in the function vectors paper. It is somewhat out of scope for our paper, which focuses on the task-detector to task-executor connection and our algorithms for SAEs.

---

> ### Comment · Reviewer_VLpn · 2024-11-25
>
> Thanks for the reply. I revisited the paper and believe that the author should consider making some improvements to the writing in future versions.

---

> > ### Author Response · Authors · 2024-11-29
> >
> > Dear Reviewer VLpn,
> >
> > Following your comment on paper writing improvements, we've made several changes to the main text:
> >
> > 1. Enhanced explanation of the TVC algorithm:
> >       - Added a detailed flowchart as Figure 2 showing key steps of the algorithm
> >       - Reworked section 3.1 to better explain the TVC algorithm
> > 2. Added $L_1$ coefficient sweep results and a brief discussion of them to section 3.1
> > 3. Strengthened evidence for task detection and execution feature distinctions:
> >       - Added max-activating examples from training distribution to the section 3.1
> >       - Added token type activation statistics tables
> >            - Included quantitative analysis showing >96% detector feature activation on output tokens and 89% executor feature activation on arrow tokens
> >       - Expanded discussion in section 3.1 on activation patterns
> > 4. Reworked section 2.1 for clarity and conciseness
> > 5. Fixed a few typographical and grammatical errors
> >
> > Additionally, we've expanded our experiments with:
> >  - $L_1$ regularization sweeps across Gemma-2 2B SAEs with varying target sparsity (L0 norms from 20-300) (Figures 13-15)
> >  - Gemma-2 9B sweeps using both 16k and 65k width SAEs (Figures 13-15)
> >
> > We hope this revision of our work, improving the writing, empirical validation, and overall presentation will motivate you to return your original score.

---

> ### Author Response · Authors · 2024-11-25
>
> Could you please tell us which sections you think require writing improvements?

---

> ### Author Response · Authors · 2024-12-02
> **Have we addressed your concerns?**
>
> Hello reviewer VLpn, **we only have 23 hours left to be able to respond to your questions** and concerns. We think that we've addressed them as detailed in our rebuttal (particularly by improving our writing), so could you let us know ASAP about further worries, or otherwise would you be open to revising your score?

---

### Official Review · Reviewer_LsDx · 2024-11-04

**Soundness:** 1
**Presentation:** 1
**Contribution:** 1
**Rating:** 3
**Confidence:** 3

**Summary:**

This work uses sparse autoencoders (SAEs) to study in-context learning (ICL). In particular, The author uses SARs to identify task-execution features with task vector cleaning algorithm and uses sparse feature circuits to find task-detection features to better understand mechanisms of ICL.

**Strengths:**

1. The author make use of sparse autoencoders (SAEs) to decompose task-vectors, which is an interesting approach to analyze task vectors.

**Weaknesses:**

1. The writing of the paper can be improved. In many places (e.g., background in Section 2.2 and 2.3, Section 3 and 4), the author uses plain text to describe the method. It would be better to include clear definitions, illustrations or formulations. The current writing is confusing.
2. Line 175 to 183 should be embedded in a Figure.
3. It seems like algorithm task vector cleaning is the main algorithm (and one of the claimed contributions) of this paper. It is better to have that in the main part of the paper rather than in the Appendix. Furthermore, Appendix C only shows an overview Figure and an example without clear definition / formulation. Since this is the main proposed method, it is better to have an algorithm description with clear step-by-step formulation (or in pseudocode). Current form is confusing.
4. Observations from line 243 to 249 lack supporting evidence (it would be better to have some visualization or quantitative metric).
5. It seems hard for me to find the main contribution of this paper. While the author lists three contributions in the introduction, the first one might not be significant enough and the last two ones are vague (what are some specific observations?)

Minor issues:
1. Full stop symbol missing at some places (e.g., line 215, 287, 356 and all equations).

**Questions:**

1. In line 153, what is "do" in "do$(a=a')$"?
2. What is the "layers of interest" in line 197?

---

> ### Author Response · Authors · 2024-11-21
> **Rebuttal by Authors**
>
> We appreciate the time and effort the reviewer has taken to engage with our work. Below, we aim to clarify the understanding of our contributions, particularly in light of the points raised regarding their significance and clarity.
>
> # Contributions
>
> ## Contribution 1: Effective Use of Sparse Autoencoders for Large-Scale Mechanistic Interpretability
>
> Our primary contribution lies in demonstrating that Sparse Autoencoders (SAEs) can be effectively used to explain mechanisms behind in-context learning (ICL) tasks in large models, such as Gemma-1 (2B), which is 10–20x larger than models typically analyzed at this depth in mechanistic interpretability research (e.g., [Wang et al., 2022](https://arxiv.org/abs/2211.00593); [Marks et al., 2024](https://arxiv.org/abs/2403.19647)). This scalability is not merely incremental; it is a significant technical achievement because it shows that causal circuit-finding methods, like SFC (sparse feature circuits), can scale to large language models (LLMs), enabling interpretability on tasks where smaller models may not produce the same behaviors. Our work shows a strong lower bound for how useful SAEs are for explaining ICL, and is successful as a proof-of-concept for interpreting ICL with SAEs
>
> ## Contribution 2: Identification of Core Bottlenecks in ICL Circuits
>
> Our second key contribution is the identification of two core bottlenecks in the ICL circuit: **task-detection** features and **task-execution** features, along with an in-depth analysis of their interactions. These features are not only causally important to the ICL circuit but also interpretable. Specifically:
>
> * We demonstrate that task-detection features and their corresponding task-execution features are causally linked via attention mechanisms and MLP layers. Figure 6 shows that by steering with negative decoder directions we can disable corresponding executor features.
> * With steering experiments in Figures 3 and 5, we show that both task-detection features and task-execution features can independently trigger task execution.
> * We also found that both types of these features are often interpretable. To give more evidence for this, we have added maximum activating examples on the SAE training data of the cleanest features we found as Appendix I.
>
> ## Contribution 3: Task Vector Cleaning (TVC)
>
> We introduced **Task Vector Cleaning (TVC)**, a novel method that decomposes task vectors into a small set of interpretable and task-relevant features, while keeping their performance intact. This approach addresses the limitations of *ITO* (inference-time optimization, [Smith et al. 2023](https://www.alignmentforum.org/posts/C5KAZQib3bzzpeyrg/progress-update-1-from-the-gdm-mech-interp-team-full-update#Replacing_SAE_Encoders_with_Inference_Time_Optimisation)), which usually degrades the performance and may miss important features.
> In response to feedback, we have expanded our analysis of TVC to include different models and SAE sizes. We compare its performance to ITO and task vectors in SAEs of two widths for Gemma 2 2B and on our own Phi-3 SAEs. This broader evaluation demonstrates that the method is effective across a range of architectures, affirming its general applicability and robustness. These results can be found in the new Appendix D.1.
>
> We are open to clarifying and discussing these contributions further in the comments, and would appreciate you reconsidering your opinion if we’ve been able to clarify our paper’s findings more clearly.
>
> # Stylistic changes
>
> We appreciate the stylistic feedback, and the new revision is adjusted accordingly. Additionally, we inserted citations for causality literature, and, for easier reading, added references to sections describing our contributions when they are mentioned.
>
> We cite papers describing the algorithms we use in detail, such as Sparse Feature Circuits in Section 2.2 and Task Vectors in Section 2.3. We also include pseudocode with concrete hyperparameters. We describe notable changes in Section 4, providing enough detail for a reader to be able to reproduce our version of the algorithm.

---

> ### Author Response · Authors · 2024-11-25
>
> Before this phase of the discussion period ends, we wanted to ask the reviewer whether we have addressed your concerns with our work?
>
> We appreciate the feedback regarding our writing and structure, and the lack of clarity in our contributions, and believe we have now addressed these concerns

---

> ### Author Response · Authors · 2024-12-02
> **Have we addressed your concerns?**
>
> Hello reviewer LsDx, **we only have 23 hours left to be able to respond to your questions** and concerns. We believe we have addressed them thoroughly in our two comments, particularly by editing our paper for clarity. Could you please let us know as soon as possible if you have any additional concerns? Alternatively, would you be open to revising your initial judgment?

---

### Author Response · Authors · 2024-11-22
**Overall Response to ICLR 2025 Reviews**

We sincerely thank all reviewers for their thoughtful feedback. We are encouraged by the reviewers' recognition of our work's key strengths:
# Recognized strengths
1. **Methodological Innovation:** “Interesting approach to analyze task vectors” (Reviewer LsDx) and a "Successful implementation of a relatively complex combination of methods (...) *effectively combines pretrained SAEs, Inference-Time Optimization (ITO), and end-to-end loss into a cohesive approach*" (Reviewer VLpn) that "*makes significant technical contributions* by developing novel methods (task vector cleaning algorithm and modifications to SFC) that enable analyzing in-context learning mechanisms in larger language models like Gemma-1 2B, which is substantially larger than models typically studied at this depth"* (Reviewer JwLJ).
2. **Showcasing Potential of SAEs:** “This work not only advances the understanding of ICL but also *showcases the potential of SAEs as powerful tools for interpretability research in LLMs*, paving the way for future studies to refine and control model behavior” and “*encourages further research and experimentation within the community* by lowering the entry barrier for other researchers”. (Reviewer VLpn)
3. **Important Research Direction:** “Understanding how ICL works is an important problem” (Reviewer BufG) and ICL is an "important capability” (Reviewer JwLJ)
4.  **Novel Insights:** “The work provides meaningful insights into how ICL works by identifying and validating two key circuit components” (Reviewer JwLJ) and “This work not only advances the understanding of ICL” (Reviewer VLpn)
5. **Clear Structure and Presentation:** “the paper is *structured in a good way*” and “the presentation using heatmap is *intuitive* and shows the effectiveness and exclusiveness of these task features” (Reviewer BufG)
6. **Rigorous Methodology:** “The methodology is *rigorous*, with *extensive ablation studies*, *careful experimental design*, and thorough validation of findings through steering experiments and causal analysis.” (Reviewer JwLJ)

# Addressing Reviewer Concerns

Reviewers raised important points about our work that we have addressed in our revision:

## Extended Implementation Details and Model Evaluation

In response to concerns about limited model selection (Reviewer VLpn, BufG) and hyperparameter details (Reviewer JwLJ, BufG), we have:

- Added $L_1$ coefficient sweep analysis in Appendix D.1 for multiple models: our Gemma-1 2B and Phi-3 Mini SAEs, and Gemmascope Gemma-2 2B SAEs (16k and 65k widths)

- Demonstrated that our method consistently reduces active features by 50-70% while maintaining task vector performance across different models and SAE widths (Appendix D.1)

- Added positive steering heatmaps for both Gemma-2 2B SAE widths, showing successful identification of task-specific executors (Appendix F.2)

## ICL literature Review

Addressing Reviewer JwLJ's request for "more direct comparisons to existing methods", we have added a detailed comparison with existing ICL analysis methods in Appendix H, demonstrating our approach's advantages and relating our findings to previous work. Additionally, we expanded the discussion on approximation errors in the Conclusion section, explaining their potential impact on our interpretations and reinforcing the validity of our findings despite these limitations.

## Improved Visualization

Following Reviewer BufG's request for "more specific characteristics that define these two kinds of features" and Reviewer LsDx's comment that "observations from line 243 to 249 lack supporting evidence" and need "visualization or quantitative metric", we have:

- Added max-activating examples for task-executing and task-detecting features (Appendix I)

- Included full unfiltered positive steering heatmaps to demonstrate the actual steering effect of task-executing features on the loss (Appendix F)

## Additional details about our method
We have addressed reviewers LsDx and BufG’s requests for additional details about TVC and SFC procedure and costs. We have:

- Included additional implementation details and hyperparameters for SAE training, circuit discovery and task vector cleaning throughout the paper to facilitate reproducibility.

- Calculated and reported the training costs for SAEs in Appendix B, demonstrating the feasibility of our approach

- Highlighted that our task vector cleanup algorithm is computationally efficient, taking approximately one minute per task on a single TPU chip.

---

### Author Response · Authors · 2024-11-29

Following up on our previous response, we thank the reviewers again for their feedback on our work.

While we have updated the paper to address the new concerns about presentation and structure, we would like to respectfully note that these concerns primarily relate to how our research is communicated rather than its underlying scientific contributions. Specifically, Reviewer BufG suggests that “the paper requires significant restructuring,” and Reviewer VLpn’s feedback is the recommendation that we “consider making some improvements to the writing in future versions.” We believe that the novelty and significance of our technical contributions – including the first successful application of SAEs to explain ICL in substantially larger models, the discovery of fundamental ICL circuit components, and the introduction of an effective new methodology – should be the primary factors in evaluating our paper's contribution to the field.

To better showcase these contributions, we have made improvements to the paper's organization and clarity:

**Writing and Structure Improvements:** We have restructured the paper to improve clarity and accessibility. Key updates include:



1. Moving the Task Vector Cleanup (TVC) algorithm flowchart from the appendix to Figure 2 in the main text, supported by expanded technical explanations that make the methodology more accessible.
2. Including a subset of the max-activating examples for task executing features and task detecting features, both found using the TVC algorithm, as Appendix I.
3. Incorporating several maximum activating examples from Appendix I into a new Figure 4, along with activation mass distribution data in Tables 1 and 2, which clearly demonstrate the distinct patterns of task detection and execution features.
4. Enhancing the contributions section to better articulate our three main advances:
    * The successful application of SAEs to explain ICL mechanisms in larger models like Gemma-1 2B, which is 10-35x larger than typical models being analyzed
    * The identification and validation of two core bottlenecks (task-detection and task-execution features) in the ICL circuit, supported by our new quantitative evidence
    * The introduction of the TVC algorithm, now demonstrated to work effectively across multiple models and SAE configurations
5. Expanding the discussion of approximation errors in the conclusion section. We want to clarify again that these limitations do not affect our core findings about the detector-executor relationship.
6. Adding a comprehensive comparison with existing ICL analysis methods in Appendix H.

**Technical and Experimental Updates:** While not central to our main contributions, we conducted additional experiments to demonstrate our approach's robustness, including evaluating Gemma 2 9B and testing various Gemma 2 2B SAE configurations with different training $L_1$ coefficients, which showed consistent performance across different L0 norms (20-300).

With the structural improvements addressing the reviewers' presentation concerns, we believe the paper now effectively communicates its significant technical achievements and novel insights into ICL mechanisms. We remain open to further discussion during the remaining review period and are committed to ensuring our work meets ICLR's high standards while advancing the field's understanding of large language models.

---

### Meta-Review · Area_Chair_FWqV · 2024-12-17

**Metareview:**

This work uses Sparse Autoencoders to study in-context learning in large language models. The study adapts methodologies to a larger model, offering new insights into ICL mechanisms. The reviewers mention the limited scope of SAE features and the models used. I strongly agree with the reviewers on this point, making it a major flaw in the proposed paper. In addition, the paper could benefit from additional case studies. Lastly, the paper does not make any more in-depth (theoretical) analysis, which would offer more solid insights, thus limiting the contribution of this work.

**Additional Comments On Reviewer Discussion:**

The reviewers were not convinced by the rebuttal - I agree with the flaws pointed out by the reviewers, regarding the novelty, the experimentation and the direction of this paper. I do agree with the reviewers and therefore I do not believe this paper is ready for acceptance. There are concerns regarding the direction and focus of this paper, which are raised in the original reviews. Thus, I do recommend to the authors to carefully consider those along with the broader idea of SAEs.

---

### Decision · Program_Chairs · 2025-01-22

Reject